# Optimal Epidemic Control as a Contextual Combinatorial Bandit with Budget

Baihan Lin
baihan.lin@columbia.edu
Columbia University
New York, NY, USA

Djallel Bouneffouf
djallel.bouneffouf@ibm.com
IBM Research
Yorktown Heights, NY, USA

## ABSTRACT

In light of the COVID-19 pandemic, it is an open challenge and critical practical problem to find a optimal way to dynamically prescribe the best policies that balance both the governmental resources and epidemic control in different countries and regions. To solve this multi-dimensional tradeoff of exploitation and exploration, we formulate this technical challenge as a contextual combinatorial bandit problem that jointly optimizes a multi-criteria reward function. Given the historical daily cases in a region and the past intervention plans in place, the agent should generate useful intervention plans that policy makers can implement in real time to minimizing both the number of daily COVID-19 cases and the stringency of the recommended interventions. We prove this concept with simulations of multiple realistic policy making scenarios and demonstrate a clear advantage in providing a pareto optimal solution in the epidemic intervention problem. [1]

## CCS CONCEPTS

• **Computing methodologies** → *Machine learning*; • **Applied computing** → **Life and medical sciences**.

## KEYWORDS

Contextual Bandit, Reinforcement Learning, Epidemic Intervention, Public Health, Policy Making

**ACM Reference Format:**
Baihan Lin and Djallel Bouneffouf. 2022. Optimal Epidemic Control as a Contextual Combinatorial Bandit with Budget. In *epiDAMIK 2022: 5th epiDAMIK ACM SIGKDD International Workshop on Epidemiology meets Data Mining and Knowledge Discovery, August 15, 2022, Washington, DC, USA.* ACM, New York, NY, USA, 8 pages.

## 1 INTRODUCTION

Consider a practical case in epidemic intervention, the prescriptor development of governmental resources and policies. In light of the global pandemic of COVID-19, many agencies have devoted considerable time and resources in finding the best solution for it. For instance, the Xprize Pandemic Response Challenge

---

[1]The data and codes to reproduce the empirical results can be accessed and reproduced at https://github.com/doerlbh/BanditZoo.

(https://www.xprize.org/challenge/pandemicresponse) is a hypothetical contest that foster the implementation of accurate and rapid prescriptions and enable ongoing improvements to the model as new interventions, treatments, and vaccinations become available. By integrating information on eco-environmental and social-economical factors, machine learning models can better forecast and prevent epidemic spread [29, 30]. The prescriptor development encompasses the rapid creation of custom, non-pharmaceutical and other intervention plan prescriptions and mitigation models to help decision-makers minimize COVID-19 infection cases while lessening economic and other negative implications of the virus. According to one example of theirs, machine-generated prescriptions may provide policymakers and public health officials with actionable locally-based, customized, and least restrictive intervention recommendations, such as mandatory masks and reduced restaurant capacity.

The *contextual bandit* problem is a variant of the extensively studied multi-armed bandit problem [5, 13, 14, 22], where at each iteration, the agent observes an $N$-dimensional *context* (*feature vector*) and uses it, along with the rewards of the arms played in the past, to decide which arm to play [2, 6, 9, 16]. The agent uses this context, along with the rewards of the arms played in the past, to choose which arm to play in the current iteration. The objective of the agent is to learn the relationship between the context and reward, in order to find the best arm-selection policy for maximizing cumulative reward over time. These online learning agents have been successfully applied to practical domains such as modeling human behaviors [24], simulating game theory [21], and even speech processing tasks [25–27].

However, in many more complicated real life problems, the sequential decision making process can involves multi-dimensional action spaces and multi-criteria optimization objectives. In another word, in the same iteration, the agent might have to make decisions simultaneously in $K$ action dimensions 1 through $K$, and within each action dimension $k$, selecting the optimal arm for that dimension. As its feedback, this combinatorial action group can sometimes yield a mixture of reward signals, such as a pair of positive reward and negative cost. This is especially true in the critical application of epidemic intervention.

With the variants of COVID-19 spreading across the globe wave after wave, it is of vital importance for us as machine learning researchers to provide intelligent solutions in various areas in epidemic control and help alleviate this looming condition that impact hundreds of millions of people. As far as we are aware, there have not been work that utilized active-learning-based method (e.g. bandits) to prescribe non-pharmaceutical intervention plans given high-dimensional cost and effect signal feedbacks. As a result, we

| Indicator | $N_j$ | values | action |
|-----------|-------|--------|--------|
| $C_1$ | 3 | (0,1,2,3) | School Closures |
| $C_2$ | 3 | (0,1,2,3) | Workplace Closures |
| $C_3$ | 2 | (0,1,2) | Cancel Public Events |
| $C_4$ | 4 | (0,1,2,3,4) | Restrictions on Gatherings |
| $C_5$ | 2 | (0,1,2) | Closing of Public Transport |
| $C_6$ | 3 | (0,1,2,3) | Stay at Home requirement |
| $C_7$ | 2 | (0,1,2) | Restrictions on internal movement |
| $C_8$ | 4 | (0,1,2,3,4) | International Travel Controls |
| $H_1$ | 2 | (0,1,2) | Public Information Campaigns |
| $H_2$ | 2 | (0,1,2) | Testing Policy |
| $H_3$ | 3 | (0,1,2,3) | Contact Tracing |
| $H_6$ | 4 | (0,1,2,3,4) | Facial Coverings |

**Table 1: Common non-pharmaceutical interventions (NPIs)**

believe that our contextual bandit solution is a timely first attempt to solve this challenging problem.

## 2 APPLICATION PROBLEM

The technical challenge of the prescriptor development can be mapped to our contextual bandit problem as follows: Based on a time sequence of the number of cases in a region and the past intervention plans in place, the agent should generate useful intervention plans that policy makers can implement. Each prescriptor agent should balance a tradeoff between two objectives: minimizing the number of daily COVID-19 cases while minimizing the stringency of the recommended interventions (as a proxy to the economic and quality-of-life costs by taking this intervention).

Since the intervention plan costs can differ across regions (e.g. closing public transportation may be more costlier in New York City than in Pittsburgh), a region-specific weights can be provided by each region government as a function to output a region-specific resource stringency given a prescription as its input. In our setting, the context to a contextual bandit will correspond to this weight vector $\mathbf{c} = \{\mathbf{c_1}, ..., \mathbf{c_k}\}$ with $k$ as its action dimensions. These action dimensions are $k$ different possible non-pharmaceutical interventions (NPIs, e.g. closing schools, lockdown, etc) are given as an input since they are used to compute the reward functions that needs to be jointly optimized, which are the scalar *stringency* metric and the scalar *number-of-cases* metric.

The action $\mathbf{a} = \{\mathbf{a_1}, ..., \mathbf{a_k}\}$ consists of severity values (0 to $N_k$) with $k$ as the index of the intervention. For instance, for the practical application of the Pandemic Response Challenge dataset, there are 12 different intervention NPIs that need to be prescribed over a 180-day period. The list of commonly used non-pharmaceutical interventions and their potential actions are given by Table 1. Each action consists of setting these variables to values within the appropriate ones for a given period of time. The effect of the number of cases can be computed in approximately 2 weeks as delayed feedback, whereas the effect of the economic consequences can be computed immediately as instant feedback.

The decision making agent will be given a time interval, the historical time series of COVID daily cases and the previous NPI actions taken, and then decide on a sequence of actions for the next day. To produce a prescriptor that minimizes both the number

of cases and the modified health containment index, we want to minimize the following 2-dimensional objective:

$$\left\{ \sum\nolimits_{\text{time}} \text{number\_cases}, \sum\nolimits_{i,\text{time}} w_i \text{ stringency}_i \right\}$$

where $w_i$ stands for weight associated with the $i^{\text{th}}$ NPI. However, it is a time-consuming process to choose an optimal intervention plan. A plan is a combination of 12 NPIs, where there are from 3 to 5 options for each intervention, resulting in a total of 7,776,000 plans per day. Combining that with the fact that standard predictor takes non-negligible time to return predictions, we see that a brute-force approach is not possible. Contextual bandits can potentially balance this tradeoff between exploration (accurate learning of the relationships between prescribed policies and their outcomes) and exploitation (early prescription of effective policies to control pandemic).

## 3 BACKGROUND

This section introduces some background concepts our approach builds upon, such as contextual bandit and contextual combinatorial bandit.

### 3.1 The Contextual Bandit problem

The contextual bandit (CB) problem has been extensively studied in the field of reinforcement learning, and a variety of solutions have been proposed [19, 20, 23]. In LINUCB [1, 10, 17, 18], Neural Bandit [4] and in linear Thompson Sampling [3, 7, 8], a linear dependency is assumed between the expected reward given the context and an action taken after observing this context; the representation space is modeled using a set of linear predictors. In [11] the authors proposed the novel framework of *contextual bandit with restricted context*, where observing the whole feature vector at each iteration is too costly or impossible for some reasons; this is related to the *budgeted learning* problem, where a learner can access only a limited number of attributes from the training set or from the test set (see for instance [12]).

Following [16], this problem is defined as follows. At each time point (iteration) $t \in \{1, ..., T\}$, a agent is presented with a *context* (*feature vector*) $\mathbf{c}(t) \in \mathbf{R}^N$ before choosing an arm $k \in A = \{1, ..., K\}$. We will denote by $C = \{C_1, ..., C_N\}$ the set of features (variables) defining the context. Let $\mathbf{r}(t) = (r_1(t), ..., r_K(t))$ denote a reward vector, where $r_k(t) \in [0, 1]$ is a reward at time $t$ associated with the arm $k \in A$. Herein, we will primarily focus on the Bernoulli bandit with binary reward, i.e. $r_k(t) \in \{0, 1\}$. Let $\pi : \mathbf{R}^N \to A$ denote a policy, mapping a context $c(t) \in R^N$ into an action $k \in A$. We assume some probability distribution $P_c(c)$ over the contexts in $C$, and a distribution of the reward, given the context and the action taken in that context. We will assume that the expected reward (with respect to the distribution $P_r(r|c, k)$) is a linear function of the context, i.e. $E[r_k(t)|\mathbf{c}(t)] = \mu_k^T \mathbf{c}(t)$, where $\mu_k$ is an unknown weight vector associated with the arm $k$; the agent's objective is to learn $\mu_k$ from the data so it can optimize its cumulative reward over time.

### 3.2 The Contextual Combinatorial Bandit

Our feature subset selection approach will build upon the *Contextual Combinatorial Bandit (CCB)* problem [28], specified as follows.

**Algorithm 1** Contextual Combinatorial Thompson Sampling with Budget (CCTSB)

---

1: **Require:** $K, N, T, C, \alpha > 0$
2: **Initialize:** $\forall k \in [K]$ and $\forall i \in [N], B_i^k := I_C, z_i^k := 0_C, \hat{\theta}_i^k := 0_C$
3: **Foreach** $t = 1, 2, ..., T$ **do**
4:    observe $c(t)$
5:    **Foreach** action dimension $k \in K$ **do**
6:      **Foreach** budget $i \in N$ **do**
7:        Sample $\tilde{\theta}_i^k$ from $\mathcal{N}(\hat{\theta}_i^k, \alpha^2 B_i^{k-1})$
8:      **End do**
9:    **End do**
10:    **Foreach** arm $k \in K$ **do**
11:      Select arm $i_k(t) := \arg\max_{i \subset [I]} c(t)^\top \tilde{\theta}_i$
12:    **End do**
13:    Observe $r(t)$ and Observe cost $s(t)$
14:    Get $r^*(t) = \frac{r(t)}{s(t)}$
15:    **Foreach** $i \in i_k(t)$
16:      $B_i^k := \lambda(t) B_i^k + c(t) c(t)^\top$
17:      $z_i^k := z_i^k + c(t) r^*(t)$
18:      $\hat{\theta}_i := B_i^{k-1} z_i$
19:    **End do**
20: **End do**

---

Each arm $k \in \{1, ..., K\}$ is associated with the corresponding variable $x_k(t) \in R$ which we assume that is sampled from a Gaussian distribution and which indicates the reward obtained when choosing the $k$-th arm at time $t$, for $t > 1$. Let us consider a constrained set of arm subsets $S \subseteq \Psi(K)$, where $\Psi(K)$ is the power set of $K$, associated with a set of variables $\{r_M(t)\}$, for all $M \in S$ and $t > 1$. Variable $r_M(t) \in R$ indicates the reward associated with selecting a subset of arms $M$ at time $t$, where $r_M(t) = f(x_k(t)), k \in M$, for some function $f(\cdot)$. The contextual combinatorial bandit setting can be viewed as a game where the agent sequentially observes a context $c$, selects subsets in $S$ and observes rewards corresponding to the selected subsets. Here we will define the reward function $f(\cdot)$ used to compute $r_M(t)$ as a sum of the outcomes of the arms in $M$, i.e. $r_M(t) = \sum_{k \in M} x_k(t)$, although one can also use nonlinear rewards. The objective of the CCB algorithm is to maximize the reward over time. We consider here a stochastic model, where the expectation of $x_k(t)$ observed for an arm $k$ is a linear function of the context, i.e. $E[x_k(t)|\mathbf{c}(t)] = \hat{\theta}_k^T \mathbf{c}(t)$, where $\hat{\theta}_k$ is an unknown weight vector (to be learned from the data) associated with the arm $k$. The outcome distribution can be different for each action arm.

## 4 CONTEXTUAL COMBINATORIAL THOMPSON SAMPLING WITH BUDGET (CCTSB)

In Algorithm 1, we introduced the Contextual Combinatorial Thompson Sampling with Budget (CCTSB). Here are some preliminaries: $K$ is the number of action dimensions; $N$ is the number of action values or arms for each action dimension $k$; $T$ is the entire time length of the learning problem; $C$ is the dimension of the context vector.

In this specific formulation for epidemic intervention, the number of action values or arms, $N$, can be different for each action dimension. For instance, there might be four levels of "School Closures", but only three levels of "Testing policies". The action values or arms are ordinal budget level, meaning that they correspond

directly to the cost (and thus also the global budget) that each arm is exhausting.

For each time step, the agent observed a context $c(t)$. Then, for each action dimension $k$ and each budget levels $i$, the agent performs a Contextual Thompson Sampling on each of this combinatorial search space, and choose the right arm or budget in each action dimensions that maximize the score $c(t)^\top \tilde{\theta}_i$. When the reward $r(t)$ and the cost $s(t)$ is revealed, the agent update its embedding with a mixed reward functions $r^*$ that coordinates between $r$ and $s$. Shown in the algorithm is one formulation, where $r^*(t) = \frac{r(t)}{s(t)}$, but depending on the application domains and empirical benefits, it can be flexibly changed into other forms, such as $r^*(t) = r(t) + \frac{\lambda}{s(t)}$ where $\lambda$ can be changed to favor one reward criteria over the other.

Having multiple instances of CCTSB with different $\lambda$ can potentially give us a pareto frontier that maximizes the both criteria in different degrees. This pareto frontier is especially important in public health policy making, as governments need to balance the tradeoff between the resource stringency and effective control of the epidemic spread.

## 5 INDEPENDENT COMBINATORIAL BANDIT (INDCOMB)

To better understand the effect of the context in this problem, we propose two additional algorithms of interest, which we call *IndComb-UCB1* and *IndComb-TS*, as the model variants that don't use the context at all. *IndComb-UCB1* and *IndComb-TS* are two combinatorial bandits that consist of $K$ independent multi-armed bandits for each of the $K$ action dimensions. The backbones we use here are Upper Confidence Bound, or *UCB1*, [15] and the Thompson Sampling, or *TS* [31], two theoretically optimal solutions in the multi-armed bandit problem. We hypothesize that in cases where the context is constant, these algorithms might perform better than CCTSB, and in cases where the context varies a lot, the CCTSB should be more effective because it utilizes these stringency-relevant information.

We wish to note that, although some algorithmic backbone we used as baselines, such as Thompson Sampling, UCB1, or Contextual Thompson Sampling, are not necessarily newly proposed by us, formulating and testing the critical and practical epidemic control problem with these state-of-the-art contextual or combinatorial bandits itself should be an important contribution by itself.

## 6 EMPIRICAL EVALUATIONS

To empirically evaluate the performance of the proposed combinatorial contextual bandit algorithm in the epidemic control problem, we created a simulation environment of different types of epidemic intervention conditions, because there is no real-life dataset available as ground truth (i.e. to obtain one, we would need to have a government perform exactly what an artificial agent recommend and record the cost and effect, which might have unexpected ethical concerns).

**Disclaimer.** Due to the serious nature of this application problem, we wish to disclaim that the use of data in the evaluation environment that we created might be a simplification from the real-world scenarios of the challenging epidemic control problem. However, as one might also note, without access to proprietary

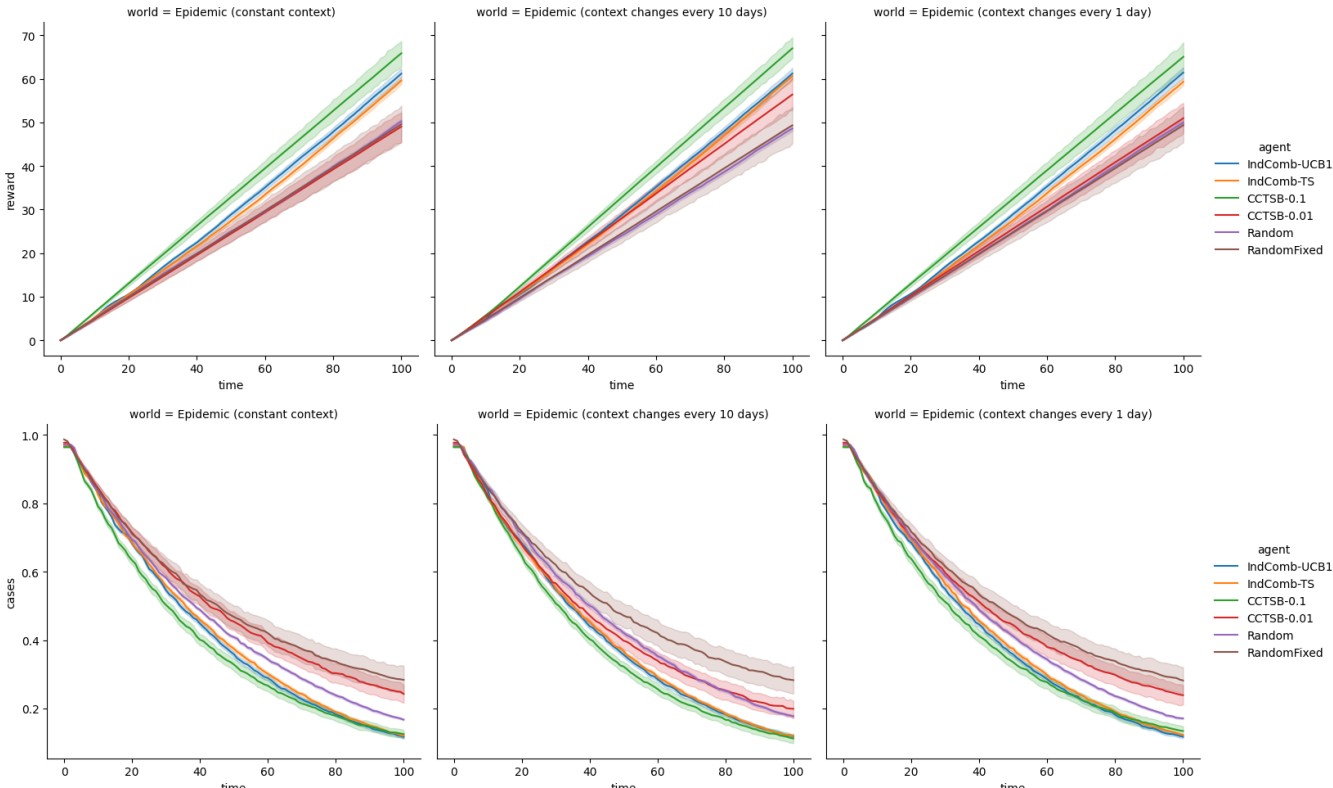

**Figure 1: Reward-driven study: CCTSB significantly reduces the number of cases and effectively controls the epidemic spread.**

governmental data it is nearly impossible to obtain an accurate estimate of the effect- and stringency-related weights to initialize our simulation environments. It is also potentially unethical to perform a controlled experiment regarding the epidemic control problem. Therefore, it is an unfortunate inherent limit that we couldn't solve in this current work. As a result, at the current stage we don't consider it as a topic of study, but focus instead on the algorithmic development to prepare for a generic epidemic intervention scenario with randomly initialized effect factors and stringency costs. We believe by randomizing the weight matrix multiple times in the simulation environment, we might provide a useful insight to generalize to real world scenarios (where each city or region has a different stringency and effect criterion).

**Simulation environments.** In our simulation, we can randomly identify $K$ action dimensions (corresponding to different non-pharmaceutical invention approaches), and randomly identify $N$ different action levels for each action dimension. For instance, we might design an epidemic control world where there are two action dimensions (i.e. $K = 2$), traffic control and school closure; traffic control can have two levels (degree 1 and degree 2, i.e. $N^{traffic} = 2$) and school closure can have three levels (all schools, all schools except universities, or all primary schools, i.e. $N^{school} = 3$). We can either consider the budget used by each intervention to be independent (each intervention approach and its value yields a fixed amount of cost regardless of other action dimensions) or combinatorial (the cost of each intervention approach and its value

depends on what the action values are in other action dimensions). In real-world, the cost for each intervention approach is usually independent from other intervention dimensions. Thus, we adopt an independent assumption for these cost weights (or stringency weights as in epidemic control terms). As introduced in the problem settings, the policy makers (i.e. our agents) have access to these stringency weights and thus can use them as contexts in this sequential decision making task.

**Stationarity priors.** In real world, the stringency weights of the government can have different stationarity priors. For instance, in certain countries and cities, the costs of many infrastructure-related intervention approaches are constant throughout the year. For instance, the cost of different levels of public information campaigns are usually constant due to the stability of the advertisement business. In some population-dense cities and countries, many intervention approaches might have highly volatile costs depending on the population flows of the districts. For instance, the traffic control might have significantly different economical costs depending on how much population remain in the cities that are working from home. Amidst the spectrum between the two extremes, there are intervention plans that has cost which varies in slower time scale, such as seasonal changes of costs. To accommodate for all three scenarios, we considered three artificial scenarios in the simulation: one that holds the stringency weights, or the context, constant throughout the learning, one that changes the stringency weights

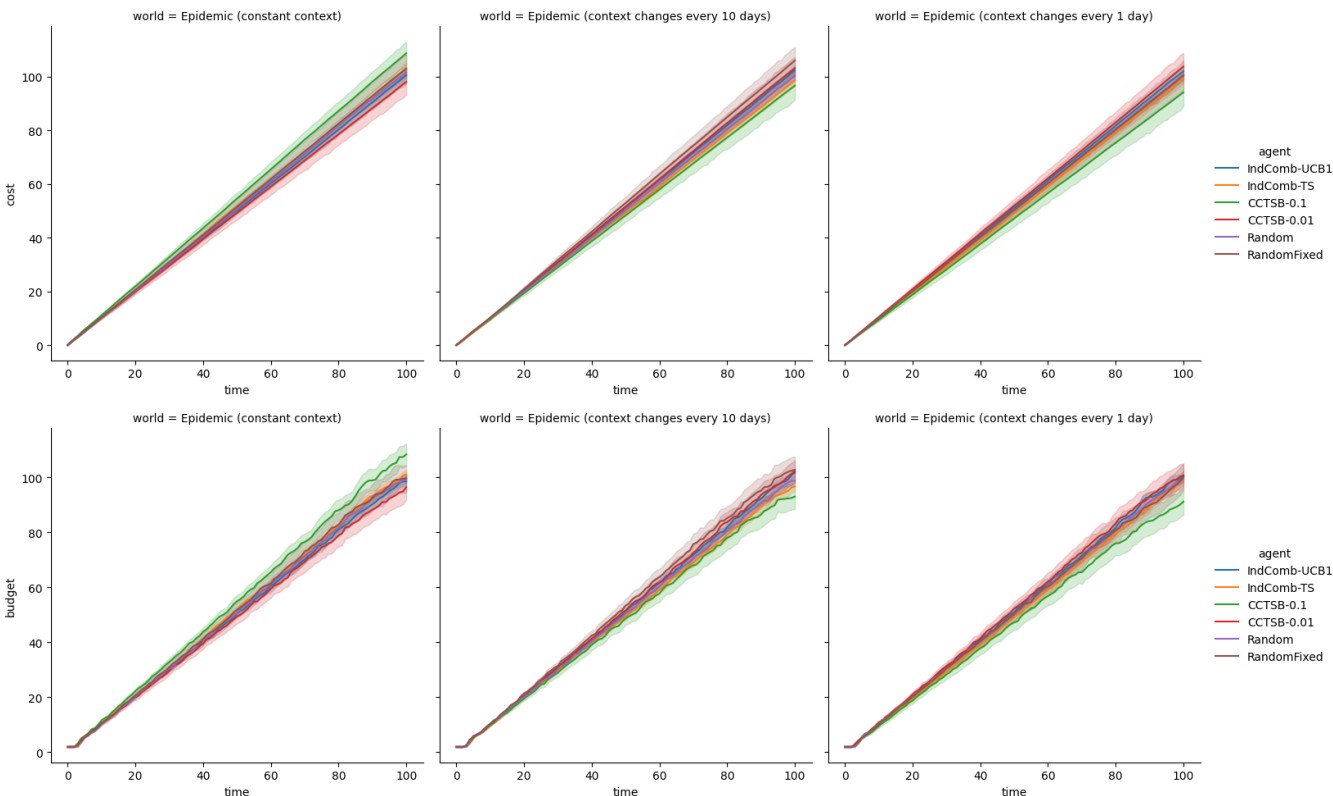

**Figure 2: Cost-driven study: CCTSB maintains a low cost consumption level and effectively constrains the resource stringency.**

every 10 decision steps, and one that changes the resource stringency weights every decision step.

**Evaluation metrics.** To evaluate the problems, we report four metrics. The reward and cost are the ones recorded by the artificial environments. To better match the realistic problem of epidemic control. We post-process these two measures to create two additional metrics corresponding to real-life measures. The "cases" is an estimate of the number of active cases that is infected by the disease, given by an exponential function of the reward: $cases = e^{-reward^*}$, where $reward^*$ is the quantile-binned normalized reward. The "budget" is a quantile-binned metric of the cost. The pareto frontier would be a curve of the number of cases over the used budget.

**Baselines.** We introduce two CCTSB agents, *CCTSB-0.1* and *CCTSB-0.01*, which either set the hyperparameter $\alpha$ to be 0.1 or 0.01. We also include IndComb-UCB1 and IndComb-TS to study the effect of contextual information in the learning. Other than the proposed CCTSB algorithm, we included two baselines. The first one is a *Random* agent, which randomly pick an action value in every action dimension in each decision step. The real-life correspondence of this type of policy is a government policy that changes every day, like a shotgun. *RandomFixed* is another baseline, where it randomly picked an action value in each action dimension at the first step, and then stick to this combinatorial intervention plan till the end. This is like a government policy that doesn't change throughout the entire epidemic period.

**Experimental setting.** In our simulation, we randomly generate multiple instances of the environments and randomly initialize multiple instances of the agents. In each world instance, we let the agents make decisions for 1000 steps and reveal the reward and cost at each step as their feedbacks. For all the evaluations, there are at least 50 random trials for each agent and we report their mean and standard errors in all figures. In the multi-objective setting, we consider the objective function to be $r^*(t) = r(t) + \frac{\lambda}{s(t)}$. In the pareto optimal evaluation, each agent was evaluated as least for 50 random trials and we report their means and standard errors.

## 6.1 Purely reward-driven bandit agents effectively control the epidemic spread

In the first scenario, we set the $\lambda$ to be 1, such that the agents are purely driven by the reward. As shown in Figure 1, comparing to the baselines, our agent *CCTSB-0.1*, is the best performing agent, significantly reducing the number of infected cases (and yielding the highest rewards), which suggests that it effectively controls the epidemic spread. We also note that if we set the hyperparameter $\alpha$ to be 0.01 (which controls for the exploration), the result is not as good. As we will see in later results, this hyperparameter can be tuned to fit different situations.

It is also worth noting that, the two combinatorial bandit that we use, *IndComb-UCB1* and *IndComb-TS*, perform relatively well, comparing to the two random baselines. This is interesting because,

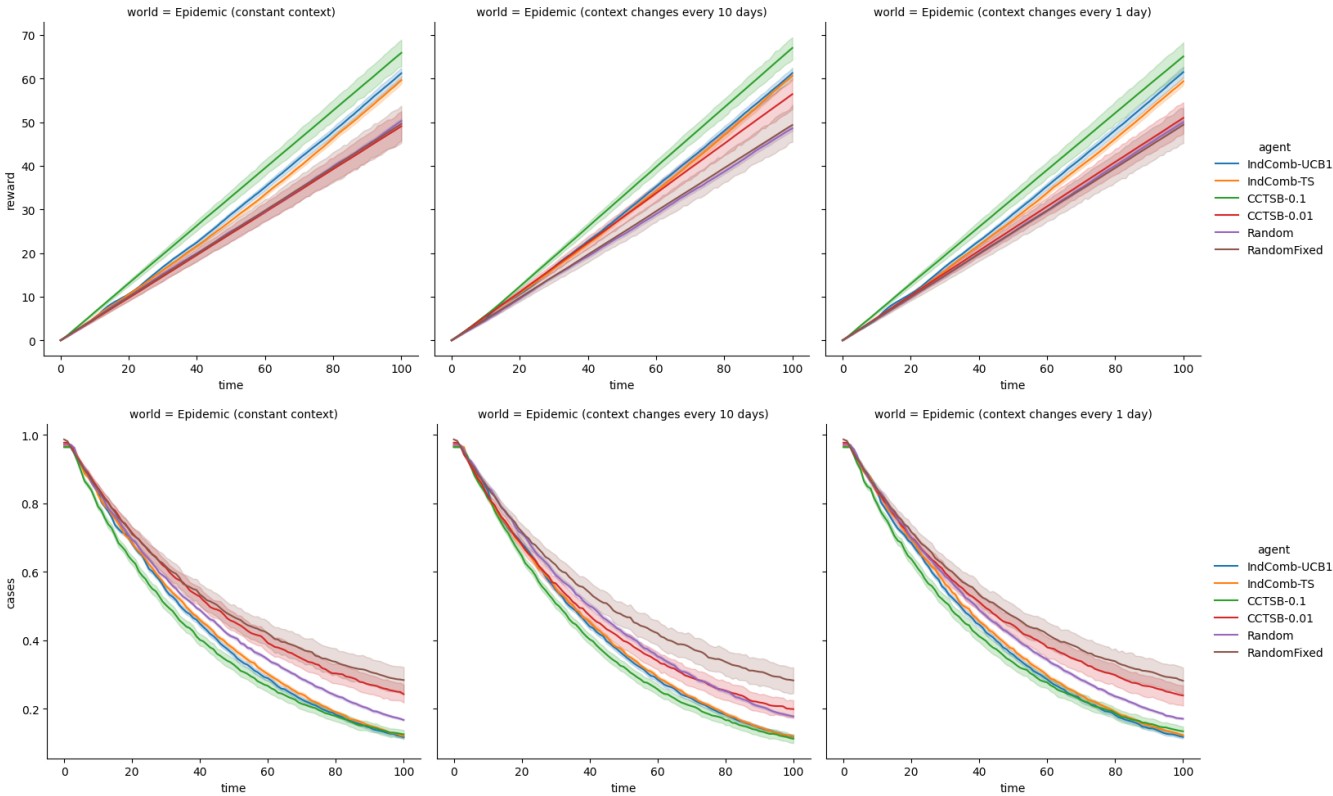

Figure 3: Cost-driven study: Despite not receiving feedback on the infection rate, CCTSB doesn't tradeoff the epidemic control.

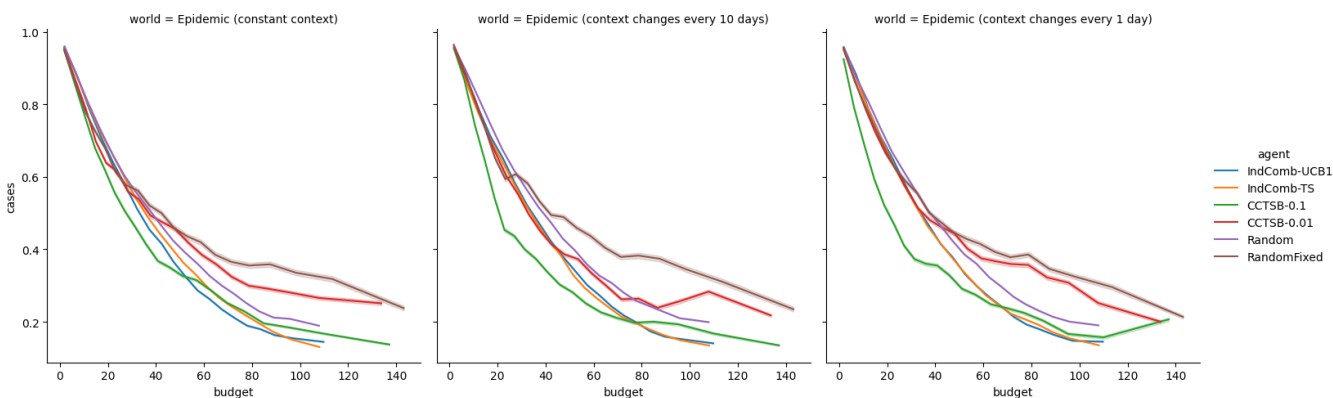

Figure 4: Pareto frontiers of the cases vs. budget in epidemic scenarios of different stationarities.

this study is not simply promoting one solution to this underexplored application, but to provide candidate methods to better understand the behaviors of different machine learning methods in response to different priors. The relatively well performance of the non-contextual bandit solutions here suggests, if the contextual information such as side knowledge are not available, policy makers should still benefit from using a bandit algorithm to optimize for their policy prescription.

## 6.2 Purely cost-driven bandit agents effectively constrain the government stringency

In the second scenario, we set the $\lambda$ to be 0, such that the agents are purely driven by the cost. As shown in Figure 2, in all three scenarios, *CCTSB* are the best performing agent that significantly reduce the budget usage, which suggests that they can effectively constrain the government stringency. We also note that *CCTSB* can be sensitive to the choice of hyperparameter $\alpha$, where a bigger

number $\alpha$ (which controls the exploration) are more suitable for more stationary condition, as in the epidemic environment with constant context.

We also observe that, even if the objective function is only dependent on the cost, the *CCTSB* does a decent job controlling the epidemic control than all the baselines (Figure 3). This suggests that the context might offer a regularization upon the learned policy, such that it doesn't converge early on an aggressive policy and can adapt when the stringency weights vary. This is an important feature for government agencies, it suggests that the recommendations by this policy might be more reversible if taken a wrong step. For instance, the policy maker might decide to focus on the stringency constraint given a temporary resource shortage, but it might not want to sacrifice the epidemic control aspect during this temporary strategy shift.

Comparing to the results in purely reward-driven cases, we observe that the combinatorial method that we propose to use, *CCTSB* is the best performing one, and each has its edges in different scenarios, while *IndComb* also performs well in many cases. We are interested in reporting them all to facilitate a more complete understanding of the problem and engage the communities to continue this line of work.

## 6.3 Pareto Optimal Solution in Epidemic Control

To obtain a pareto frontier for the agents in the epidemic simulation, we run the above evaluations with different values of $\lambda$ ranging from (0,0.25,0.5,0.75,1). Then we quantile-binned the metrics for each agent and plot out their average and standard errors.

As shown in Figure 4, our proposed algorithms yield the pareto optimal frontier, that every intervention plan extracted on its curve will minimize both the number of infected cases in a given day and the resource budget on the government. We also observe that when the context is constant or slow changing, and if we have some leeway in the budget side, we can get slightly better infection control with the *IndComb*'s recommendations. This suggests that for the governments whose stringency features are constant or slow changing, a combinatorial bandit might suffice as the policy making engine to yield a pareto optimal solution for epidemic control. However, in other realistic conditions where the stringency weights vary every now and then, the CCTSB is more effective in extracting these useful information in decision making. We recommend the policy making agencies take into account the specific stationarity of their stringency constraints to choose the best problem-specific solution between our multi-objective combinatorial bandit framework for epidemic control.

## 7 CONCLUSION

In summary, we introduce a series of combinatorial bandit algorithms for the epidemic control problem, including a novel formulation of contextual combinatorial bandits that continually learns to prescribe a multidimensional action groups that maximizes a multiobjective reward functions. It provides a pareto optimal solution to a practical healthcare problem of epidemic intervention, such that given historical policy prescription and epidemic progression, the agent can continually prescribe future intervention plans in

different domains that minimizes both the pandemic daily cases and the government resource stringency.

Active and reinforcement learning has been successfully applied to different applications including but not limited to game playing and computer vision, but the important application of prescribing epidemic intervention policies happens to be mostly a field of blank. Therefore, we believe our application contributes to the field in a nontrivial way.

We demonstrate in multiple simulation environments of epidemic worlds under real-world assumptions that the proposed combinatorial contextual bandit can effective balance the tradeoff between the epidemic control and resource stringency, and offer the optimal pareto frontier of the epidemic intervention problem. We believe this machine learning solution can help stakeholders like governmental officials to create interpretable intervention policies that control the global pandemic in a timely and efficient manner.

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
