# OpenReview forum: "Optimal Epidemic Control as a Contextual Combinatorial Bandit with Budget"
_ACM.org/SIGKDD/2022/Workshop/epiDAMIK — KDD 2022 Workshop epiDAMIK Poster_

### Official Review · Reviewer_JEps · 2022-06-25
**The paper needs more elaboration to deliver its methods and results clearly.**

**Rating:** 2
**Confidence:** 3

**Review:**

### Summary

The problem proposed by this paper aims to find an optimal intervention strategy that considers the trade-off between infected cases and the cost of the intervention policies. This paper proposes various combinatorial bandit algorithms for generating time-varying epidemic control strategies. The algorithm incorporates contextual information and previous policy reward history to learn hidden bandit function weights. Experiments on several simulated environments validate that the proposed algorithms outperform random selection baselines and non-contextual bandit algorithms.

### Weakness

It is hard to understand the algorithms proposed in this paper. For example, in Section 4 and 5, this paper refers to previous works, e.g., Contextual Thompson Sampling, Upper Confidence Bound, and Thompson Sampling, to describe their algorithms. It would be better to elaborate on the algorithm descriptions or provide more related work discussion.

As for experiments, more information could be included in the experimental setup, such as simulation environments. For example, how are the functions in the simulation environments defined? This necessary information could help readers to interpret the results.

The experimental results show the reward, budget, and case curves of different epidemic control strategies. It would be more helpful to include the exact definitions of the y-axis (e.g., cases). Moreover, in Figure 4, the results are confusing, since the curve of CCTSB-0.1 is higher than IndComb-TS in the second half of the simulation period. It would be better to explain the reason for these results.

The writing of the paper could be improved, especially the problem formulation and algorithm designs. Providing formal definitions of the problem and notation table would help readers understand the paper better.

---

### Official Review · Reviewer_YJ8s · 2022-06-25
**Interesting idea of using Contextuals Bandits in Epidemic Control**

**Rating:** 3
**Confidence:** 4

**Review:**

-The paper attempts to show that Contextual bandits can potentially balance the tradeoff between accurate learning of the relationships between prescribed policies and their outcomes and early prescription of effective policies to control a pandemic with two objectives: minimizing the number of daily COVID-19 cases and minimizing the stringency of the recommended interventions. This is done using well established algorithms such as Thompson Sampling, UCB1, and Contextual Thompson Sampling (aside from using these algorithms there is not much theoretical contribution). Although the experiments are based on simulations and not real world data, the idea of using contextual bandits for epidemic control is interesting. With respect to language, the paper needs some revisions.
The following are some comments that might help improve the paper.



Some comments:
- In the abstract, the authors say "tradeoff of exploitation and exploration" which is disconnected from what they were saying before that and is not clear what they are referring to. Later exploitation and exploration are explained in the Application Problem section. The abstract should be revised to be clearer.
- Regarding policy making using covid data, the authors say in the abstract that " We prove this concept with simulations of multiple realistic policy making scenarios and demonstrate a clear advantage in providing a pareto optimal solution in the epidemic intervention problem." I am confused about what they are trying to prove. Whatever that would be, "simulations" are not proofs.
- The authors claim that " there have not been work that utilized active-learning-based method (e.g. bandits) to prescribe non-pharmaceutical intervention plans given high-dimensional cost and effect signal feedbacks. As a result, we believe that our contextual bandit solution is a timely first attempt to solve this challenging problem. " Doing some simple search, I'm not sure if that is true and related work should be cited.
- The first half of the first paragraph of Section 3.1 is not necessary or can be moved to the introduction.
- How is the hyperparameter $\alpha$ chosen (to be 0.1 or 0.01)?


Some minor comments:

ABSTRACT
- to find a optimal way to dynamically: a ----> an
2 APPLICATION PROBLEM
-are given as an input since:  an input ----> inputs
where 𝑤𝑖 stands for: remove the space
3.1 The Contextual Bandit problem
a agent is presented with: a ----> an
5 INDEPENDENT COMBINATORIAL BANDIT (INDCOMB)
-We wish to note that: there is nothing wrong with we note that
7 CONCLUSION
-combinatorial contextual bandit can effective balance the: effective ---->effectively

---

### Official Review · Reviewer_pzhN · 2022-06-27
**Promising work but its applicability is still to be proven.**

**Rating:** 3
**Confidence:** 3

**Review:**

This work aimed at demonstrating the efficiency of contextual combinatorial bandits with budget in generating intervention plans that policy-makers could in turn use to minimize of a pandemic such as COVID-19. The technique and the experiments designed to assess the effects of the former are quite clearly presented, although a substantial number of typos are present. The introduction of multiple baselines in order to compare the proposed model and the level of detail provided about the experiments is relevant. Here are several improvement ideas:
1. The assumption that costs for each arm are independent is questionable. For instance, if stay-at-home policies are applied, closing public transportation might have a negligeable effect, far smaller than if the population hadn’t been confined.
2. In the ‘Experimental setting paragraph’, it is claimed that simulations are run for 1000 time steps, but all plots have a time scale between 0 and 100.
3. In 6.1 and 6.2, the value for $\lambda$ seem to have been exchanged ($\lambda=0$ would cancel the effect of cost in the mixed reward function).
4. As noted, it is not evident that such a model could be used by actual policy-makers in a real-world situation. The public might be skeptical about accepting the decisions of an algorithm, especially knowing that it relies partly on exploration, which by definition explores options with unknown or little-known outcomes, or knowing that it could change policies often.
5. Figures are not colorblind-friendly and not particularly easy to read, it could have been useful to provide zooms on certain parts.
6. References [14] and [15] are duplicates.
It would be interesting to obtain more information about the applicability of this model, for instance with regard to the variations of stringency weights.